# Effects of Elevated CO$_2$ Concentration and Temperature on the Growth and Photosynthetic Characteristics of *Populus simonii* × *P. nigra* '1307' Leaves

**Nan Xu** [1,†]**, Junnan Ding** [1,†]🆔**, Tianyi Zhang** [1]**, Juexian Dong** [1]**, Yuan Wang** [1] **and Xuechen Yang** [2,*]🆔

[1] Key Laboratory of Heilongjiang Province for Cold-Regions Wetlands Ecology and Environment Research, Harbin University, Harbin 150086, China; xunan0451@126.com (N.X.); ding.junnan@163.com (J.D.); zty136038174@163.com (T.Z.); 13078783263@163.com (J.D.); wy20012021@163.com (Y.W.)

[2] State Key Laboratory of Black Soils Conservation and Utilization, Northeast Institute of Geography and Agroecology, Chinese Academy of Sciences, Harbin 150081, China

[*] Correspondence: yangxuechen@iga.ac.cn

[†] These authors contributed equally to this work.

**Abstract:** This study aimed to establish the effects of elevated CO$_2$ concentration and temperature on the photosystem II (PSII) performance and photosynthetic characteristics of *Populus simonii* × *P. nigra* '1307' leaves. Different CO$_2$ concentrations (400 and 800 μmol·mol$^{-1}$) and temperatures (room temperature and room temperature +2 °C) were set in artificial climate change simulation and control chambers, and the rapid chlorophyll fluorescence induction kinetics curve (OJIP curve) of *Populus simonii* × *P. nigra* '1307' was determined. The generated OJIP curve was used to analyze the change characteristics in photosynthetic performance. The results revealed that under elevated temperature conditions, the concentrations of chlorophyll *a*, *b* in *Populus simonii* × *P. nigra* '1307' leaves were significantly increased. At the same time, there were no significant changes in the chlorophyll concentration under the superimposed effect of elevated CO$_2$ concentration and temperature. The PSII comprehensive performance index (PI$_{ABS}$) of *Populus simonii* × *P. nigra* '1307' was significantly inhibited under elevated temperatures due to the increased closure degree ($V_j$) of the PSII reaction center and the damage of the receptor side. This reduced the electron transfer capacity per unit reaction center (ET$_o$/RC) and unit cross-sectional area, which decreased the quantum yield of the electron transfer. Under the elevated CO$_2$ concentration, ET$_o$/RC was also inhibited. Still, PI$_{ABS}$ was enhanced owing to the increased number of active PSII per unit area and the low reduction rate of the primary quinone receptor (Q$_A$). Under the superimposed effect of the two factors, the electron transfer performance of the donor and receptor sides of PSII was improved compared to the treatments only subjected to elevated temperature; thus, PI$_{ABS}$ was not significantly reduced compared to the control. Therefore, the continuous increase in temperature by 2 °C significantly inhibits the electron transfer capacity of the photosynthetic system of *Populus simonii* × *P. nigra* '1307' leaves. On the other hand, an increase in CO$_2$ concentration expands the PSII reaction center, while enhancing the electron transfer capacity of the donor and receptor sides, which alleviates the photosynthetic inhibition caused by the elevated temperature.

**Keywords:** *Populus simonii* × *P. nigra* '1307'; elevated CO$_2$ concentration; increased temperature; photosynthesis; rapid chlorophyll fluorescence

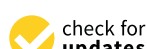



## 1. Introduction

The vegetation types in natural ecosystems worldwide are shaped by the different environmental temperatures, moisture, soil characteristics and biological factors. Since the industrial revolution, human activities have greatly increased the emission of greenhouse gases, resulting in a series of environmental problems, including the greenhouse effect, nitrogen deposition, and acid rain. These challenges directly or indirectly

influence the plant–environmental balance, leading to forest recession, wetland degradation, and crop yield reduction. However, plants have developed a series of response mechanisms to environmental changes following their evolution, altering vegetation productivity. As a result, the scientific community has assessed and predicted the current and potential future environmental changes, providing a strong theoretical basis for studying plant responses to various environmental changes. For example, the IPCC WGI Fifth Assessment Report pointed out that the atmospheric $CO_2$ concentration would double the $CO_2$ concentration before the industrial revolution by 2030, and the global temperature would rise by 1.5–4.5 °C [1].

Global warming has increased the evapotranspiration rate in terrestrial ecosystems, affecting the global precipitation patterns, which directly impact plants during water and high-temperature stresses. An increase in environmental temperatures significantly advances the spring phenology but postpones the autumn phenology, while prolonging the growth season of trees [2]. Forests are crucial in maintaining the global carbon–water balance [3], while serving as terrestrial carbon sinks. While an increased atmospheric $CO_2$ concentration elevates the atmospheric temperature, the high $CO_2$ absorption in forest ecosystems may alleviate the greenhouse effect. Additionally, the elevation of the atmospheric $CO_2$ concentration increases the photosynthetic rate of plants and improves the carbon fixation capacity of the vegetation [4]. The carbon sequestration processes, including photosynthesis, tree growth, and litter decomposition, are the main prerequisites for forest ecosystems becoming carbon sinks. For example, the forest ecosystem in North America could offset 20–40% of greenhouse gas emissions, while the terrestrial vegetation carbon sink in China could counteract 14–16% of greenhouse gas emissions [5]. Therefore, exploring the impacts of climate factors on the carbon sequestration process in forests is of great significance in mitigating the greenhouse effect.

The continuous increase in atmospheric $CO_2$ concentration and temperature is a major environmental factor affecting plant growth and development under the changing climate globally. Precisely, the leaf photosynthetic system responses are sensitive to climate change. For example, in *Glycine max* and rice, increased $CO_2$ concentration significantly increases the PSII actual light quantum yield and photochemical quenching [6,7], promoting the speed of light energy capture and transformation efficiency into chemical energy by chloroplast, which enhances the photosynthetic capacity. In contrast, high temperatures inhibit the hydrolysis of the PSII donor side, limiting the photoelectron transport capacity and potential photochemical efficiency of plants [8]. For example, high-temperature stress reduces the photochemical efficiency, actual quantum yield, and electron transfer rate in soybean leaves, reducing their photosynthetic capacity [9]. Several studies have explored the impact of a single factor (increased $CO_2$ concentration or temperature) on the developmental process, aboveground growth, underground growth, and photosynthesis of forest vegetation. However, there are limited reports on the interaction between these two factors. Therefore, to accurately predict the impact of future global climate changes on forest ecosystems, it is necessary to better understand the responses and adaptations of the forest vegetation to the simultaneous increase in atmospheric $CO_2$ concentration and temperature.

*Populus* spp. is a model species for studying woody plants due to their relatively simple genome, fast growth, strong adaptation, and wide distribution. They play an important role in water and soil conservation, carbon sequestration, and ecological restoration. By 2015, the total area occupied by artificial *Populus* spp. forest in China exceeded 100 million mu, ranking first in the world and greatly contributing to forest engineering in China. Global *Populus* spp. studies are continuously promoting the construction of artificial *Populus* spp. forests consisting of well-adapted varieties to meet social and economic demands while protecting the environment. In this study, an artificial climate chamber was used to control $CO_2$ concentration and temperature to conduct a two-factor interactive pot experiment, and different $CO_2$ concentration and temperature levels were set to study the effects of increasing $CO_2$ concentration and temperature on photosynthesis and rapid chlorophyll fluorescence characteristics of *Populus* spp. trees. It is urgent to study the effects

of increasing $CO_2$ concentration and temperature on light energy, capture, transfer, and carbon assimilation. To understand the response and adaptation mechanism of *Populus* spp.'s photosynthetic system to $CO_2$ concentration and temperature increase. The findings of this study will provide important theoretical and practical insights into the adaptation mechanism of trees to environmental changes for the development of high-quality artificial *Populus* spp. forests in northern China.

## 2. Materials and Methods

### 2.1. Experimental Materials

*Populus* spp. is the main naturally distributed tree species in China and is widely distributed in the northern regions (Harbin China). The greenhouse effect and increased temperature affect the growth and biomass allocation of *Populus* spp. Therefore, it is important to establish the influence of continuous environmental changes on the physiological characteristics of *Populus* spp. In this study, potted *Populus* spp. trees were used. The response mechanism of *Populus* spp. to environmental changes was revealed from the growth and physiological aspects by setting the artificial climate chamber to simulate an environment with the greenhouse effect and temperature rise.

### 2.2. Experimental Design

At the beginning of April 2022, healthy *Populus simonii* × *P. nigra* '1307' stem cuttings were planted in outdoor awning pots with local topsoil consisting of sandy loam soil. The physical and chemical properties of the soil used were pH = 7.98, a total nitrogen content of 0.835 g/kg, available nitrogen of 43.88 mg/kg, available phosphorus of 6.78 mg/kg, available potassium of 107.2 mg/kg, organic matter of 11.94 g/kg, and total salt of 0.06%. The pots consisted of plastic buckets with a diameter of 50 cm and a depth of 50 cm. Before planting, a layer of little pebbles was placed at the bottom of each pot, and then each pot was filled with 10 kg of air-dried soil. Two layers of newspapers were placed between the pebbles and the air-dried soil for separation. In addition, a PVC pipe was inserted through the air-dried soil into the pebbles for watering to control the soil moisture content.

After the cuttings were planted, the plants were maintained for one month, and the seedlings with consistent growth were selected at the beginning of May for subsequent experiments. The experiments were conducted in four artificial climate change simulations and control chambers. The chambers had a steel structure covered with tempered glass and natural lighting with a light transmittance of about 85%. Each chamber was 3.5 m high with an area of 5 m × 7 m. An automatic control module controlled the air temperature and $CO_2$ concentration. The air temperature in the control chamber was consistent with the outdoor temperature. The target temperature was achieved using a high-power air conditioner controlled by a master computer based on the differences between indoor and outdoor temperatures. The indoor air was collected, and the $CO_2$ concentration was measured through the control system using the $CO_2$ sensor in the chamber. The data were then transferred to the master computer. The control program controlled the opening and closing of the solenoid valve in each chamber. Thus, the $CO_2$ content in each chamber was maintained at the target concentration. At the same time, circulating fans were also installed in the chambers to ensure a uniform indoor temperature and $CO_2$ concentration. This system also performed the monitoring of the air humidity and soil humidity. The experimental settings for the four chambers were as follows: the control chamber was consistent with the environmental conditions (CK, $400 \pm 20$ $\mu mol \cdot mol^{-1}$ of $CO_2$ concentration, and the temperature was consistent with the outside), the high-$CO_2$ concentration chamber (eC, $800 \pm 20$ $\mu mol \cdot mol^{-1}$ of $CO_2$ concentration, and the temperature was consistent with the outside), the high-temperature chamber (eT, $400 \pm 20$ $\mu mol \cdot mol^{-1}$ of $CO_2$ concentration, and the temperature was $2 \pm 0.5$ °C higher than the environment), and interactive air chamber (eC + eT, $800 \pm 20$ $\mu mol \cdot mol^{-1}$ of $CO_2$ concentration, the temperature was $2 \pm 0.5$ °C higher than the environment). Twenty plants with the same growth trend were selected for

further analysis and each experiment treated five plants. The experimental treatment lasted four months, and all experimental data were collected at the beginning of September.

### 2.3. Determination of the Tree Growth Status

Plant growth: 20 plants with the same growth trend were selected for further analysis. The plant height and ground diameter were measured using tape and a vernier calliper, respectively. A scanner (DS-60000, EPSON, Suwa, Japan) was used to scan *Populus* spp.'s leaves, and the leaf area coefficients of 0.75 were obtained through image processing. The leaf area (leaf length × leaf width × 0.75) was also calculated.

Each measurement (five separate samples) was repeated five times.

The relative water content of leaves: Fresh leaves were soaked in distilled water for 12 h, and the saturated fresh weight was measured. Next, the leaf samples were desiccated at 105 °C for 0.5 h, and then dried at 75 °C to a constant weight. Their dry weight was measured using an electronic balance with an accuracy of 0.0001 g. The relative water content (RWC) of the leaves was then calculated as follows: RWC = saturated leaf weight − fresh leaf weight/fresh leaf weight − dry leaf weight. In addition, the leaf mass per area (LMA) was also determined as follows: LMA = leaf dry weight/leaf area. Each measurement (five separate samples) was repeated five times.

Photosynthetic pigment measurement: Healthy functional leaves were selected, and 1 g of the fresh leaves were excised using scissors. The excised leaf samples were placed in 25 mL of 80% acetone and maintained away from light for 24 h. Colorimetry was performed with a spectrophotometer at the wavelength of 665 and 649 nm, and the contents of chlorophyll *a* and chlorophyll *b* were calculated. According to the OD values of each wavelength, the concentrations of chlorophyll a, b, and carotenoids ($mg \cdot L^{-1}$) were calculated, and the contents of each pigment ($mg \cdot g^{-1}$ fresh weight) were calculated according to the obtained concentrations. Pigment content in leaves (mg/g) = pigment concentration (mg/L) × total volume of extract (mL) × dilution ratio/sample mass (g). Each measurement (five separate samples) was repeated five times.

$C_a$ = 13.95D665 − 6.88D649
$C_b$ = 24.96D649 − 7.32D665
$C_{x.c}$ = (1000D470 − 2.05Ca − 114.8Cb)/245

Plant biomass: The leaves, stems, and roots were separated, and their fresh weight was weighed separately. The samples were desiccated, then dried to a constant weight, and their dry weight was measured. The dried leaves were ground into powder using a pestle and mortar and then passed through a 1 mm sieve. The leaf nitrogen content was estimated using the Kjeldahl method, and the photosynthetic nitrogen utilization efficiency (PNUE, $\mu mol\ mol^{-1}s^{-1}$) was calculated using the photosynthetic parameters and leaf nitrogen content: PNUE = $P_n/N_{area}$, where $N_{area}$ = $N_{mass}$ × LMA [10].

Determination of leaf photosynthetic physiological indicators.

Healthy and mature functional leaves were selected to assess the leaf photosynthetic physiological indicators. The light and $CO_2$ response curves were generated using a portable photosynthetic apparatus (Ciras-3, PP Systems, Amesbury, MA, USA). The flow rate was set at 500 $\mu mol \cdot s^{-1}$, and the leaf temperature was 28 °C. The light radiation intensity in the generation of the light response curve was set at 1800, 1600, 1400, 1200, 1000, 800, 600, 400, 300, 200, 160, 120, 90, 60, 40, 20, and 0 $\mu mol\ m^{-2}s^{-1}$. The light radiation intensities were fitted by the non-rectangular hyperbolic Ye model, and the initial quantum efficiency, the maximum net photosynthetic rate ($P_{max}$), light compensation point, light saturation point, and dark respiration rate were generated. The $CO_2$ concentration was set at 2000, 1800, 1600, 1400, 1200, 1000, 800, 600, 400, 200, 100, and 50 $\mu mol\ mol^{-1}$. The $CO_2$ compensation point and photorespiration rate (Rp) were generated. Instantaneous photosynthesis was measured under a light intensity of 1800 $\mu mol\ m^{-2}s^{-1}$, and gas exchange parameters, including the net photosynthetic rate ($P_n$), intercellular $CO_2$ ($C_i$), stomatal conductance ($G_s$), and transpiration rate ($T_r$), were determined. Furthermore, instanta-

neous water use efficiency (WUE) was calculated as WUE = $P_n/T_r$. Each measurement (five separate samples) was repeated five times.

Chlorophyll fluorescence measurement: The leaves were dark-adapted for 30 min, then clamped using a fluorometer (FMS-2, Hansa, Peterborough, UK). Subsequently, the minimum fluorescence ($F_o$) and the maximum fluorescence ($F_m$) were measured. Next, the photochemical light was turned on, and the steady-state fluorescence ($F_s$), the maximum fluorescence ($F'_m$) and the minimum fluorescence ($F'_o$) under light adaptation were measured. In addition, the maximum photochemical efficiency ($F_v/F_m$), primary light energy capture efficiency ($\phi_{PSII}$), non-photochemical quenching coefficient (NPQ), and photochemical quenching coefficient ($q_P$) of PSII were calculated. Finally, the rapid fluorescence induction kinetic curve was generated to estimate the photosynthetic electron transfer rate (ETR). Each measurement (five separate samples) was repeated five times.

Determination of rapid chlorophyll fluorescence kinetic curve (OJIP): The OJIP curve was generated using the Handy PEA Plant Efficiency Analyzer. The OJIP curve was induced using 3000 $\mu mol \cdot m^{-2} s^{-1}$ pulsed red light. Each treatment was repeated five times, and the OJIP curve was plotted by averaging the fluorescence intensities of the five replicates. The photosynthetic performance index ($PI_{ABS}$) based on the absorbed light energy was obtained by JIP-test analysis of the OJIP curve. ABS/RC: absorption flux per RC; $TR_o/RC$: trapped energy flux per RC (at $t = 0$); $ET_o/RC$: electron transport flux per RC (at $t = 0$); $DI_o/RC$: dissipated energy flux per RC (at $t = 0$); ABS/$CS_m$: light energy absorbed per unit area; $TR_o/CS_m$: energy captured per unit area used to reduce $Q_A$ (when $t = tF_M$); $ET_o/CS_m$: quantum yield of electron transport per unit area (when $t = tF_M$); $DI_o/CS_m$: energy dissipation per unit area (when $t = tF_M$); $RC/CS_m$: number of reaction centers per unit area (when $t = tF_M$); $W_k (=V_k/V_j)$: the K phase is relatively variable fluorescence; $\Psi_{Po}$: maximum photochemical efficiency ($t = 0$); $\Psi_o$: 2 ms light the degree of openness of the active reaction center; $\Psi_{Eo}$: quantum yield for electron transport (at $t = 0$); $\Psi_{Do}$: quantum ratio for dissipation (when $t = 0$); $M_o$: the maximum rate at which $Q_A$ is restored; $V_j$: the degree of shutdown of the active reaction center at 2 ms, that is, the degree to which $Q_A$ is restored. Each measurement (five separate samples) was repeated five times (Table S1).

### 2.4. Data Analysis

The data obtained in the experiment were obtained using Excel 2016 (Microsoft Corp., Redmond, WA, USA) and Sigma Plot 12.5 Software (Systat Software Inc., San Jose, CA, USA) was used for data sorting and chart drawing, SPSS 23.0 was used for statistical analysis, and the comparison of each treatment was carried out using the least significant difference (LSD) method; the data used in the chart are average ± scale precision difference.

## 3. Results

### 3.1. Effect of $CO_2$ Concentration and Temperature on Populus simonii × P. nigra '1307' Leaves Biomass

Both high $CO_2$ concentration and temperature significantly increased the LMA. However, the $CO_2$ concentration had a greater impact on LMA. Specifically, treatment with $CO_2$ alone had a more significant impact or interaction effect on LMA than the treatment subjected to elevated temperature. The elevated $CO_2$ concentration reduced the leaf area, while the temperature rise did not affect the leaf area. The RWC of leaves determines the water retention capacity of plants under environmental stress. However, high $CO_2$ concentration and temperature significantly increased the RWC, with the largest RWC value recorded under interactive conditions (Table 1).

**Table 1.** Effects of elevated $CO_2$ and temperature on growth changes.

| | CK | eC | eT | eC + eT |
|---|---|---|---|---|
| Leaf dry weight (g) | 11.48 ± 1.26 a | 11.96 ± 1.34 a | 11.12 ± 2.32 a | 13.50 ± 2.33 a |
| Stem dry weight (g) | 45.23 ± 4.02 a | 33.56 ± 2.34 b | 34.03 ± 2.34 a | 33.12 ± 3.78 a |
| Root dry weight (g) | 29.76 ± 5.23 c | 53.18 ± 9.56 a | 34.25 ± 8.15 bc | 47.22 ± 8.23 a |
| Plant Height (cm) | 171.32 ± 3.13 a | 163.44 ± 5.28 ab | 158.89 ± 5.97 bc | 153962 ± 3.13 c |
| Ground diameter (cm) | 0.84 ± 0.02 a | 0.85 ± 0.32 a | 0.85 ± 0.31 a | 0.84 ± 0.13 a |
| RWC (%) | 95.28 ± 2.54 c | 97.27 ± 5.23 ab | 98.45 ± 4.21 a | 96.48 ± 0.06 a |
| LMA (gm$^{-2}$) | 75.34 ± 6.23 c | 113.23 ± 6.51 a | 108.25 ± 5.12 a | 92.33 ± 4.45 b |

Note: Data in the figure are the mean ± SE; values followed by different lowercase letters indicate a significant difference ($p < 0.05$).

Interactions between increased $CO_2$ concentration and temperature resulted in a slight decline in plant height but had no significant impact on the ground diameter and dry weight of the leaves. With the increase in $CO_2$ concentration, the dry weight of roots was significantly increased, while stems were significantly decreased. Overall, the increase in atmospheric $CO_2$ concentration and temperature significantly affected the dry matter allocation in *Populus simonii* × *P. nigra* '1307' plants, while the increase in temperature had no significant effect. Compared to the controls, the underground root biomass under the different treatments was increased to varying degrees. At the same time, the mass fraction in the aboveground stem and the plant heights were significantly decreased. Additionally, the lowest plant height was recorded under the interaction of high temperature and $CO_2$ concentration, implying that the greenhouse effect significantly affects the growth of *Populus simonii* × *P. nigra* '1307'.

*3.2. Effects of $CO_2$ Concentration and Temperature on Chlorophyll Parameters on Populus simonii × P. nigra '1307' Leaves*

An increase in the atmospheric $CO_2$ concentration had no significant effect on the chlorophyll "a" and "b" concentrations. However, the increase in temperature significantly increased the chlorophyll *a* and *b* concentrations (Figure 1).

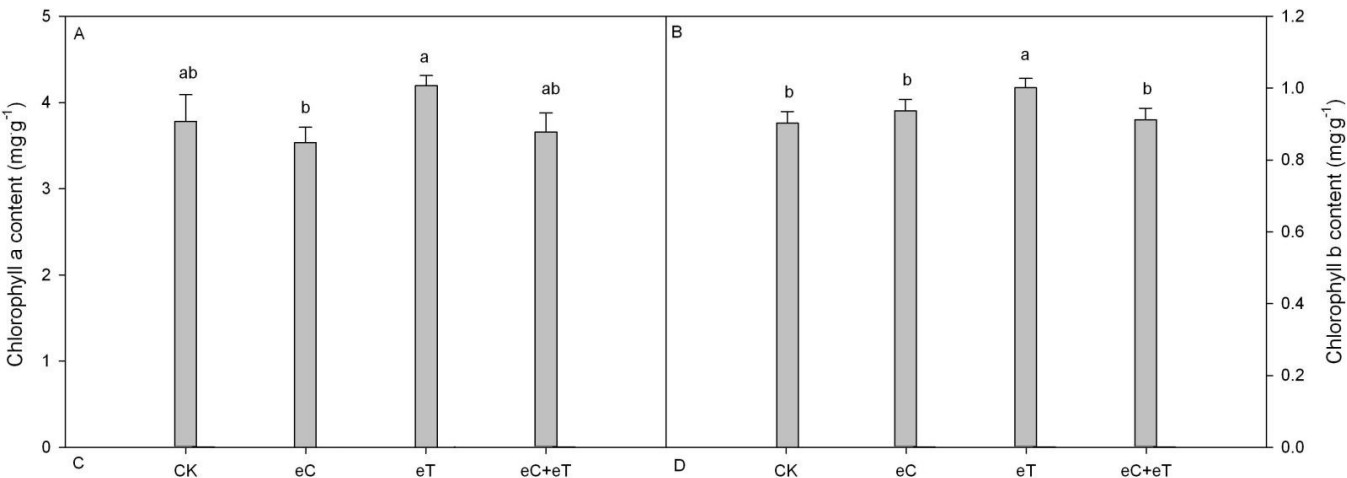

**Figure 1.** Effects of $CO_2$ concentration and temperature on chlorophyll parameters on *Populus simonii* × *P. nigra* '1307' leaves. (**A**) chlorophyll a content; (**B**) chlorophyll b content; CK: Under 400 ± 20 μmol·mol$^{-1}$ $CO_2$ concentrations; CK+ $CO_2$: CK Under 800 ± 20 μmol·mol$^{-1}$ $CO_2$ concentrations; eT: 400 ± 20 μmol·mol$^{-1}$ of $CO_2$ concentration, and the temperature was 2 ± 0.5 °C higher than the environment; eC + eT: 800 ± 20 μmol·mol$^{-1}$ of $CO_2$ concentration, the temperature was 2 ± 0.5 °C higher than the environment. Note: Data in the figure are the mean ± SE; values followed by different lowercase letters indicate a significant difference ($p < 0.05$).

### 3.3. $CO_2$ Concentration and Temperature Effects on Photosynthetic Parameters and Dry Matter Allocation in Populus simonii × P. nigra '1307' Functional Leaves

The increase in atmospheric $CO_2$ concentration and temperature had different effects on the photosynthetic physiology of *Populus simonii* × *P. nigra* '1307'. The increase in $CO_2$ concentration promoted photosynthesis, while the increase in temperature had no significant effect on photosynthesis. Thus, under the $CO_2$–temperature interaction conditions, the $CO_2$ concentration had a leading role in promoting photosynthesis. At the same time, the temperature rise had no significant impact on the gas exchange parameters in *Populus simonii* × *P. nigra* '1307' compared to the control. With the increase in $CO_2$ concentration, WUE, $P_n$, $P_{max}$, and $C_i$ were significantly increased, while $G_s$ and $T_r$ were decreased. Specifically, $G$s was decreased by 45% under high $CO_2$ concentration compared to the control. Overall, the gas exchange parameters under the interaction effect of the two factors were almost consistent with those under high $CO_2$ concentration. However, PNUE under the interaction effect of high $CO_2$ concentration and the elevated temperature was significantly higher than in the other treatments (Table 2).

**Table 2.** Effects of elevated $CO_2$ and temperature on leaf A-Ci characteristics.

|  | CK | eC | eC + eT | eT |
|---|---|---|---|---|
| $P_{max}$ ($\mu$mol m$^{-2}$s$^{-1}$) | 15.48 ± 1.06 b | 17.22 ± 1.23 b | 19.22 ± 1.82 a | 15.01 ± 0.86 b |
| $P_n$ ($\mu$mol m$^{-2}$s$^{-1}$) | 18.23 ± 0.46 b | 21.13 ± 0.34 a | 22.45 ± 1.26 a | 16.23 ± 0.83 b |
| $G_s$ (mol m$^{-2}$s$^{-1}$) | 1.44 ± 0.36 a | 0.84 ± 0.24 c | 1.01 ± 0.32 bc | 1.32 ± 0.54 ab |
| $T_r$ (mmol m$^{-2}$s$^{-1}$) | 14.45 ± 0.88 b | 12.24 ± 1.34 a | 13.82 ± 0.12 a | 14.13 ± 0.24 b |
| $C_i$ ($\mu$mol mol$^{-1}$) | 385.28 ± 15.36 a | 552.13 ± 23.52 a | 549.65 ± 18.52 b | 398.45 ± 12.21 a |
| WUE ($\mu$mol mmol$^{-1}$) | 0.97 ± 0.03 b | 1.37 ± 0.23 a | 1.67 ± 0.23 a | 0.97 ± 0.03 b |
| PNUE ($\mu$mol mol$^{-1}$s$^{-1}$) | 113.27 ± 8.23 b | 108.24 ± 6.24 b | 132.56 ± 22.35 a | 103.56 ± 2.22 b |

Note: Data in the figure are the mean ± SE; values followed by different lowercase letters indicate a significant difference ($p < 0.05$).

### 3.4. Effects of $CO_2$ Concentration and Temperature on Chlorophyll Fluorescence Parameters of Functional Leaves

The ETR and $F_v/F_m$ were not significantly different under high $CO_2$ concentration compared to the control (Figure 2A,B). In addition, the photochemical ability of PSII was not significantly improved with increased $CO_2$ concentration. However, all parameters were consistent under temperature and $CO_2$ interactive conditions and following treatment with $CO_2$ alone, which was also not significantly different from the control. Meanwhile, NPQ was significantly decreased, while the qP was significantly increased, implying that the ability of leaves to absorb and utilize light energy was significantly increased (Figure 2C,D). The carboxylation efficiency (CE) and photorespiration rate (Rp) were significantly improved by temperature increment, and the consumption of photosynthetic products was increased (Figure 2E,F). However, the increase in temperature had no significant effect on chlorophyll fluorescence parameters and exhibited the same effects as those recorded under the interaction of elevated temperature and high $CO_2$ concentration (Figure 2).

### 3.5. Effects of High $CO_2$ Concentration and Elevated Temperature on the Activity of PSII Reaction Center in Populus simonii × P. nigra '1307' Leaves

The eT treatment did not significantly alter the unit reaction center parameters (ABS/RC, TR$_o$/RC, ET$_o$/RC, and DI$_o$/RC) compared to the control. However, ABS/RC, TR$_o$/RC, ET$_o$/RC, and DI$_o$/RC were significantly reduced following treatment with eC and eC + eT compared to the control. Nonetheless, when eT treatment was applied, ET$_o$/RC was significantly reduced by 12.36% (Figure 3).

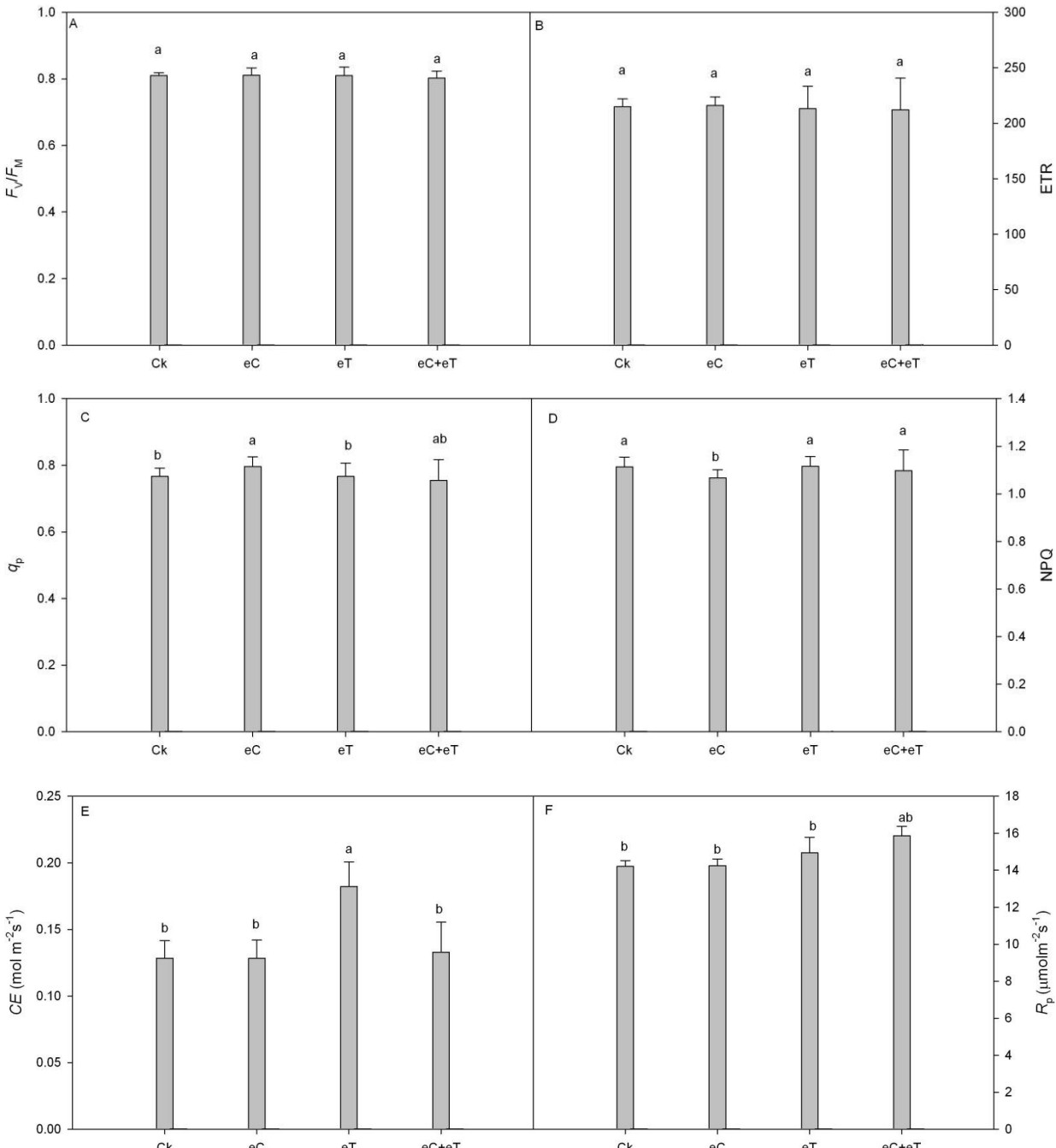

**Figure 2.** Effects of $CO_2$ concentration and temperature on chlorophyll parameters on *Populus simonii* × *P. nigra* '1307' leaves. (**A**) the maximum photochemical efficiency; (**B**) the photosynthetic electron transfer rate; (**C**) photochemical quenching coefficient; (**D**) non-photochemical quenching coefficient; (**E**) The carboxylation efficiency; (**F**) photorespiration rate; CK: Under $400 \pm 20$ μmol·mol$^{-1}$ $CO_2$ concentrations; CK + $CO_2$: CK Under $800 \pm 20$ μmol·mol$^{-1}$ $CO_2$ concentrations; eT: $400 \pm 20$ μmol·mol$^{-1}$ of $CO_2$ concentration, and the temperature was $2 \pm 0.5$ °C higher than the environment; eC + eT: $800 \pm 20$ μmol·mol$^{-1}$ of $CO_2$ concentration, the temperature was $2 \pm 0.5$ °C higher than the environment. Note: Data in the figure are the mean $\pm$ SE; values followed by different lowercase letters indicate a significant difference ($p < 0.05$).

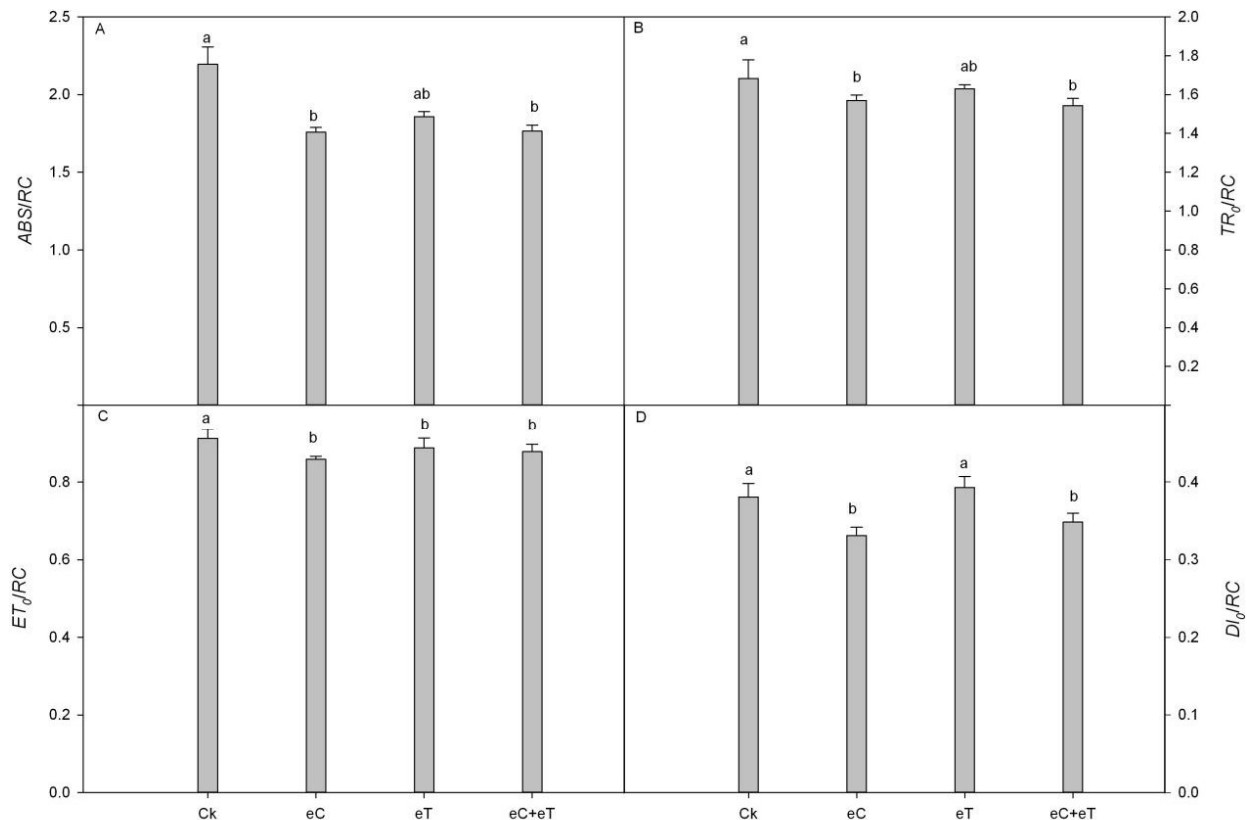

**Figure 3.** Effects of elevated $CO_2$ concentration and temperature on PSII active parameters from OJIP induction curves. (**A**) absorption flux per RC; (**B**) TRo/RC: trapped energy flux per RC (at t = 0); (**C**) ETo/RC: electron transport flux per RC (at t = 0); (**D**) DIo/RC: dissipated energy flux per RC (at t = 0)CK: Under $400 \pm 20$ μmol·mol$-1$ CO2 concentrations; CK + CO2: CK Under $800 \pm 20$ μmol·mol$^{-1}$ $CO_2$ concentrations; eT: $400 \pm 20$ μmol·mol$^{-1}$ of $CO_2$ concentration, and the temperature was $2 \pm 0.5$ °C higher than the environment; eC + eT: $800 \pm 20$ μmol·mol$^{-1}$ of $CO_2$ concentration, the temperature was $2 \pm 0.5$ °C higher than the environment. Note: Data in the figure are the mean $\pm$ SE; values followed by different lowercase letters indicate a significant difference ($p < 0.05$).

*3.6. Effects of Elevated $CO_2$ Concentration and Temperature on Donor and Receptor Sides Parameters of PSII in Populus simonii $\times$ P. nigra '1307'*

The $W_k$ was reduced by 13.96 and 14.12% under eC and eC + eT treatments compared to the control, respectively (Table 3), implying that the increase in temperature had no significant impact on the function of the PSII donor side. In contrast, the increase in $CO_2$ concentration improved the performance of the PSII donor side under both temperatures.

**Table 3.** Effects of elevated $CO_2$ concentration and temperature on PSII donor and acceptor sides parameters from OJIP induction curves.

| | $W_k$ | $V_j$ | $M_o$ | $\Psi_{Po}$ | $\Psi_o$ | $\Psi_{Eo}$ | $\Psi_{Do}$ |
|---|---|---|---|---|---|---|---|
| CK | $0.48 \pm 0.03$ a | $0..44 \pm 0.03$ a | $0.68 \pm 0.02$ ab | $0.86 \pm 0.04$ ab | $0.67 \pm 0.02$ a | $0.56 \pm 0.04$ a | $0.19 \pm 0.04$ b |
| eC | $0.42 \pm 0.04$ b | $0.47 \pm 0.03$ ab | $0.63 \pm 0.03$ ab | $0.87 \pm 0.05$ a | $0.64 \pm 0.01$ ab | $0.54 \pm 0.03$ a | $0.19 \pm 0.02$ b |
| eT | $0.46 \pm 0.02$ ab | $0.49 \pm 0.04$ a | $0.73 \pm 0.03$ a | $0.85 \pm 0.03$ b | $0.62 \pm 0.02$ b | $0.52 \pm 0.02$ b | $0.20 \pm 0.02$ a |
| eC + eT | $0.42 \pm 0.05$ b | $0.16 \pm 0.02$ a | $0.61 \pm 0.05$ b | $0.86 \pm 0.06$ ab | $0.65 \pm 0.03$ a | $0.55 \pm 0.03$ a | $0.20 \pm 0.01$ a |

Note: Data in the figure are the mean $\pm$ SE; values followed by different lowercase letters indicate a significant difference ($p < 0.05$).

Under eT treatment, $M_o$, $\Psi_{Po}$, and $\Psi_{Do}$ were not significantly different compared to the control, $V_j$ was significantly increased by 13.32%, while $\Psi_o$ and $\Psi_{Eo}$ were decreased by 7.91 and 9.06%, respectively. Under eC + eT treatment, $V_j$, $\Psi_{Po}$, $\Psi_o$, $\Psi_{Eo}$, and $\Psi_{Do}$ were

not significantly changed. These results revealed that a single treatment with increased temperature increases the closure degree of the PSII reaction center in leaves, leading to massive accumulation of $Q_A^-$ on the receptor side, thereby reducing the electron transfer efficiency. In contrast, the increase in $CO_2$ concentration under the two temperature conditions increased the opening degree of the PSII reaction center to different degrees, reducing the reduction rate of $Q_A$ while promoting the electron transfer on the receptor side.

### 3.7. Effect of Elevated $CO_2$ Concentration and Temperature on the PSII Performance Indicators in Populus simonii × P. nigra '1307'

With the eT treatment, $PI_{ABS}$ significantly decreased by 22.36% compared to the control, while no significant effects were found under eC and eC + eT treatment (Figure 4).

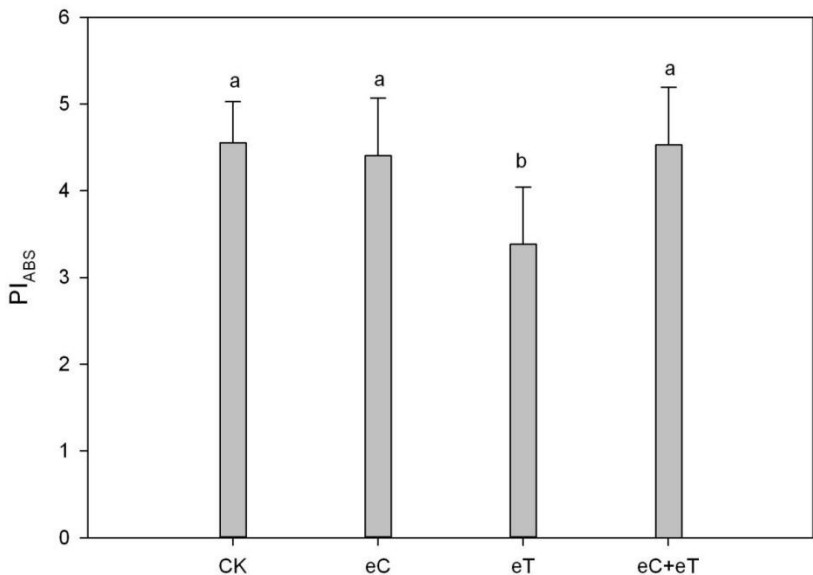

**Figure 4.** Effects of elevated $CO_2$ concentration and temperature on $PI_{ABS}$ of PSII of *Populus simonii* × *P. nigra* 1307 leaves. CK: Under $400 \pm 20$ $\mu mol \cdot mol^{-1}$ $CO_2$ concentrations; CK + $CO_2$: CK Under $800 \pm 20$ $\mu mol \cdot mol^{-1}$ $CO_2$ concentrations; eT: $400 \pm 20$ $\mu mol \cdot mol^{-1}$ of $CO_2$ concentration, and the temperature was $2 \pm 0.5$ °C higher than the environment; eC + eT: $800 \pm 20$ $\mu mol \cdot mol^{-1}$ of $CO_2$ concentration, the temperature was $2 \pm 0.5$ °C higher than the environment. Note: Data in the figure are the mean $\pm$ SE; values followed by different lowercase letters indicate a significant difference ($p < 0.05$).

## 4. Discussion

Photosynthesis is important in plant growth and development by regulating the material cycle and energy flow in various ecosystems. The photosynthetic capacity of plants is affected by the plant genetic and external environmental factors. This study assessed the impact of climate factors, including $CO_2$ and temperature, on *Populus simonii* × *P. nigra* '1307' seedlings that are of interest to international scientific communities. In addition, the photosynthetic physiology-related responses to changes in $CO_2$ and temperature in *Populus simonii* × *P. nigra* '1307' are discussed.

### 4.1. Effects of Increased Atmospheric $CO_2$ and Elevated Temperatures on Leaf Growth and Chlorophyll Content of Populus simonii × P. nigra '1307' Leaves

Leaf functional characteristics mainly include specific leaf weight, leaf relative water content, chlorophyll content, and other factors. Changes in leaf internal biomass affect leaf photosynthetic physiology, growth, and development, and play an important role in plant defense against changes in external environments. Specific leaf weight (LMA) refers to the dry weight of leaves per unit leaf area, which reflects the accumulation and transfer of plant materials and represents the photosynthetic carbon assimilation ability of plants [11]. Previous studies have shown that specific leaf weight is positively correlated

with leaf photosynthetic capacity and water use efficiency [12]. In this study, LMA increased significantly under high $CO_2$ concentration, temperature increase, and interaction, and photosynthetic rate and water use efficiency also increased significantly under high $CO_2$ concentration, which was consistent with previous research results. However, there was no significant change between the two under temperature increase, and material accumulation and biomass transfer to leaves increased as temperature increased. It may be an adaptive strategy of dry matter distribution in plants responding to environmental stress.

Photosynthetic pigment content of leaves is an important factor affecting photosynthetic capacity. Functional leaves mainly capture light energy through chlorophyll for photosynthesis, which can reflect the strength of plant photosynthetic capacity to a certain extent. Photosystem I (PSI) and photosystem II (PSII) are both composed of the core complex of chlorophyll a and the antenna complex containing chlorophyll a and chlorophyll b. Chlorophyll b has the function of absorbing and transmitting light energy and regulating the stability of photosynthetic mechanisms. When plants adapt to environmental changes, they generally adjust the chlorophyll a/b ratio to adapt to different physiological and environmental needs [13]. In this study, the increase in $CO_2$ concentration had no significant effect on photosynthetic pigments, and the increase in temperature increased chlorophyll content, but chlorophyll a/b decreased and carotenoid content also decreased significantly.

### 4.2. Effects of Increased Atmospheric $CO_2$ and Elevated Temperatures on Photosynthetic Physiology of Populus simonii × P. nigra '1307' Leaves

Previous studies revealed that an increase in atmospheric $CO_2$ improves the photosynthetic capacity of plants to varying degrees. Among the summarized direct impact of the foreign simulated increase in atmospheric $CO_2$ concentration on plants, the net photosynthetic rate was improved to varying degrees [14]. A study on the photosynthetic physiological characteristics of soybean under high $CO_2$ concentration revealed that the increase in $CO_2$ concentration improved the primary reaction efficiency of the photosynthetic reaction center in C3 plants [15]. In this study, an enhanced photosynthetic rate was not achieved by improving the primary reaction efficiency of PSII under increased $CO_2$ concentration. For example, $F_v/F_m$ and ETR were not significantly different from the control. However, the light energy captured and light energy utilization efficiency of the light reaction center were significantly increased. This is due to the increased absorbed light energy allocated to photosynthesis following increased $CO_2$ concentration, which serves as a photosynthetic substrate. The increase in atmospheric $CO_2$ concentration with an increase in atmospheric temperature causes the greenhouse effect.

However, the increase in temperature alone in this study had no significant impact on the photosynthetic capacity of *Populus simonii × P. nigra* '1307'. Similar results were observed using *P. tremuloides*, where no changes in the photosynthetic rate were reported under elevated temperature conditions alone [16]. In addition, Long et al. assessed the impact of increased temperature on plants and revealed that the increase in Rp is faster than the net photosynthetic rate under increased temperature conditions [17]. This implies that plants have not developed an adaptive strategy to increase the temperature in the short term. Therefore, future studies should explore the impact of prolonged exposure to high temperatures to unravel the response mechanism of *Populus simonii × P. nigra* '1307' to elevated temperatures. Still, the response in photosynthetic physiology under the interaction between high $CO_2$ concentration and the elevated temperature was consistent with that under high $CO_2$ treatment, implying that the $CO_2$ concentration has a greater impact on photosynthesis than the temperature in *Populus simonii × P. nigra* '1307'.

Stomata are sensitive to environmental and plant physiological factors, such as temperature, atmospheric $CO_2$ concentration, respiration rate, water potential, and soil moisture [18]. They are channels for $CO_2$ and water exchange between plants and the atmosphere [19]. Changes in the stomatal structural characteristics affect physiological functions, including transpiration and photosynthesis in leaves, which regulates the adaptation of plants to the external environment [20]. Plant gas exchange is controlled by adjusting the

stomatal opening and closing degrees to regulate the intercellular $CO_2$ concentration and by adjusting the RuBP activity to adapt to environmental changes [21]. These two processes interact, allowing the gas exchange process to regulate the plant response to complex environmental changes. The $G_s$ is an important indicator of the balance and water and the $CO_2$ cycle between plants and the atmosphere. It affects the diffusion of atmospheric $CO_2$ from the air to the chloroplast photosynthetic reaction center [22]. Given the influence of atmospheric $CO_2$ concentration on the leaves, a high $CO_2$ concentration may reduce the $G_s$ on the leaves [23]. In this study, an increase in the atmospheric $CO_2$ significantly reduced the $G_s$, implying that a high concentration of $CO_2$ enhanced the photosynthetic rate of plants. As a result, the concentration of photosynthate polysaccharide in guard cells and the water potential of cells were increased, leading to water absorption, which expanded the guard cells and subsequently the stomatal closure [22], which is consistent with the findings of a previous study [22]. In addition, the increased temperature had no significant effect on $G_s$. However, the $G_s$ under the interaction between $CO_2$ concentration and temperature were significantly lower than the control, which is consistent with the response under high $CO_2$ concentration. These results imply that the change in $G_s$ is related to photosynthesis and not the photosynthetic rate, which was not significantly affected by an increased temperature.

Correlation between the $T_r$ and environmental factors ranks light intensity, air temperature, relative humidity, and atmospheric water potential from the highest to the lowest [24]. The greenhouse effect due to human activities alters the rate of transpiration in forests, affecting the balance of regional water resources. In this study, a high $CO_2$ concentration significantly reduced the $T_r$, while an increase in temperature had no significant impact on $T_r$. Effects from the interaction between $CO_2$ concentration and temperature on $T_r$ were consistent with the response under $CO_2$ treatment. Transpiration in plants is comprehensively affected by plant characteristics and environmental factors [25]. Thus, the relationship between photosynthesis and transpiration rate is regulated by the $G_s$ [26], since when $G_s$ is significantly reduced under high $CO_2$ concentration, the transpiration rate is also reduced.

The WUE is an important indicator of water consumption per unit production of dry matter in plants. It reflects the water consumption characteristics during the physiological and ecological processes of vegetation [27], which reveals the adaptation strategy of terrestrial ecosystems to global change. The WUE is determined by the coupling effect of photosynthetic productivity and the transpiration effect. Climate changes characterized by elevated $CO_2$ concentration and global warming significantly affect the WUE in plants by altering plant productivity and transpiration [28]. A FACE experiment using 11 models to simulate the impact of $CO_2$ concentration on forests revealed a positive correlation between $CO_2$ concentration and WUE, where the WUE increased with the increase in atmospheric $CO_2$ concentration [29]. A study on the response of *Pinus koraiensis* seedlings to high $CO_2$ concentration also revealed that an increase in $CO_2$ concentration significantly reduces $G_s$ and $T_r$ but increases $P_n$ and WUE [30]. This is consistent with the results in this study, which revealed that the WUE is significantly increased under high $CO_2$ concentration and the interaction of high $CO_2$ concentration and elevated temperatures than the control. Although the increase in temperature had no significant effect on the WUE, it was related to the net photosynthetic and transpiration rates. However, there were no significant changes in the net photosynthetic and transpiration rates; hence, the WUE was not significantly altered either.

The PNUE is the photosynthetic rate per unit nitrogen content in the leaf, which is positively correlated with the plant nitrogen use efficiency; thus, it is an effective indicator of the nitrogen input for photosynthesis. One study showed the PNUE and photosynthetic rate in nearly 60 different functional plants and revealed that PNUE is significantly positively and negatively correlated with the photosynthetic rate and LMA, respectively [29]. In this study, the PNUE and $P_{max}$ were significantly higher in the interaction of high $CO_2$ concentration and elevated temperatures than in other treatments. However, a high $CO_2$ concentration and increased temperature independently had no significant impact on the

PNUE. The inconsistent PNUE responses under high $CO_2$ concentration might be related to LMA and the leaf nitrogen content.

*4.3. Enhanced Energy Conversion by Isncreased $CO_2$ Concentration in the PSII Reaction Center*

The OJIP curve is used to analyze the specific activity parameters in the PSII unit reaction center and unit cross-sectional reaction center, which characterize the change characteristics of antenna pigment size and electron transfer capacity under abiotic stress [31,32]. In this study, the continuous increase in $CO_2$ concentration significantly inhibited ABS/RC, $TR_o$/RC, and $ET_o$/RC in the leaf unit reaction center. These results are consistent with those of japonica rice under elevated $CO_2$ concentrations [33]. In our study, $ET_o$/RC was simultaneously low under elevated temperatures but was not significantly improved (Figure 2). However, under the superimposed effect of high $CO_2$ concentration and elevated temperatures, the inhibition of electron transfer per unit reaction center caused by high temperature was alleviated due to the increased number of reaction centers induced by the increased $CO_2$ concentration; hence, the activity of reaction centers per unit area was unchanged.

*4.4. Enhanced Performance of PSII Donor and Receptor by Increased $CO_2$ Concentration and Temperature*

An analysis of $W_k$, $V_j$, $M_o$, and $\Psi_{Eo}$ reveals the cause of the PSII damage and decline of electron transfer capacity [32,34]. Previous studies revealed that high-temperature stress inhibited the ETR of the PSII donor side in *Prunus armeniaca* and tomato and of the receptor side in *Populus simonii* × *P. nigra* '1307' [9,35,36]. In this study, the elevated temperatures did not significantly increase the $W_k$ of the oxygen-evolving complex (OEC) on the PSII donor side in *Populus simonii* × *P. nigra* '1307' leaves. However, they increased the $V_j$ of the active reaction center, which damaged the PSII receptor side, and reduced the $\Psi_{Eo}$, thereby increasing the inhibition of $ET_o$/RC in the reaction center. The reduction rate ($M_o$) is a specific marker of blocked electron transfer from $Q_A$ to the secondary quinone acceptor on the PSII receptor side, which characterizes the $Q_A$ reduction rate [32,35]. In this study, increasing the $CO_2$ concentration alone and the simultaneous increase in temperature and $CO_2$ concentration improved the OEC performance on the PSII donor side and reduced the $M_o$ of $Q_A$. As a result, ETR ($\Psi_{Eo}$) was unaltered compared to the control, which protected the electron receptor side from damage.

*4.5. Role of the Increased $CO_2$ Concentration on the Enhanced PSII Comprehensive Performance*

$PI_{ABS}$ is a comprehensive performance index of PSII based on light energy absorption, which accurately reflects the overall state of the PSII reaction center [37]. In cyanobacteria, $PI_{ABS}$ is decreased under high-temperature stress [38,39], mainly due to the damage of the PSII donor side, the over-reduction in the receptor side, and the decrease in energy transferred from the unit center of PSII. In this study, $PI_{ABS}$ was significantly decreased due to the increase in temperature but significantly increased under increased $CO_2$ concentration. In addition, the superimposed effect of the two factors significantly increased $PI_{ABS}$ compared to the increased temperature alone, which is beneficial for the normal progress of leaf photosynthesis ($P_{max}$) in the late developmental stage of *Populus simonii* × *P. nigra* '1307'. Several studies have revealed that under the superimposed effect of elevated temperature and increased $CO_2$ concentration, the substance accumulation and yield at the later stage of crop growth are significantly inhibited due to the shortening of the crop development period under high temperatures [38,40], pollen abortion, and the reduction in grain filling [19,41]. Thus, the $PI_{ABS}$ of the PSII reaction center in unit time is not the reason for this inhibition.

**5. Conclusions**

The increase in atmospheric $CO_2$ concentration mainly improves the photosynthetic characteristics of *Populus simonii* × *P. nigra* '1307' leaves by increasing the photosynthetic substrates and regulating the leaf anatomical structure. The photosynthetic pigments are

not significantly changed, but the stomatal density is decreased, instantaneously improving WUE. The increase in $CO_2$ concentration significantly increases the dry matter accumulation in *Populus simonii* $\times$ *P. nigra* '1307' and the biomass allocation to the roots. The stomatal density in *Populus simonii* $\times$ *P. nigra* '1307' leaves is significantly increased to dissipate heat when the air temperature is increased by 2 °C. Although the carboxylation rate and chlorophyll *b* content are increased, the photosynthetic characteristics are not significantly influenced. In addition, an elevated Rp increases the photosynthate consumption and reduces the dry matter accumulation in *Populus simonii* $\times$ *P. nigra* '1307'. The changes in photosynthetic characteristics and biomass allocation in *Populus simonii* $\times$ *P. nigra* '1307' under the interaction of the two factors were between those of single factor treatment, implying that high temperature partially offsets the promotion effect of high $CO_2$ concentration on photosynthetic physiology and the growth of *Populus simonii* $\times$ *P. nigra* '1307'.

Although the increase in $CO_2$ concentration inhibits the light energy absorption and capture ability of the PSII unit reaction center, the electron transfer capacity and the comprehensive performance of PSII are mainly maintained by increasing the number of reaction centers and reducing the reduction degree of the receptor side. Although a temperature rise by 2 °C does not significantly change the number of PSII reaction centers, their opening degree is restrained, significantly reducing the electron transfer capacity and the comprehensive performance of PSII. In addition, an increase in $CO_2$ concentration alleviates the inhibition of increased temperature on the number of PSII reaction centers, the electron transfer performance of the donor/receptor side, and the comprehensive performance of PSII.

**Supplementary Materials:** The following supporting information can be downloaded at: https://www.mdpi.com/article/10.3390/f14112156/s1, Table S1: Formulae and terms used in the analysis of the O-J-I-P fluorescence induction dynamics curve.

**Author Contributions:** Conceptualization, N.X. and J.D. (Junnan Ding); methodology, N.X.; formal analysis, Y.W.; investigation, T.Z. and J.D. (Juexian Dong); resources, X.Y.; writing—original draft, N.X. and J.D. (Junnan Ding); writing—review and editing, N.X. and X.Y. All authors have read and agreed to the published version of the manuscript.

**Funding:** This research was supported by the Heilongjiang Province Natural Science Foundation (LH2022C053), Natural Science Foundation of Jilin Province (YDZJ202201ZYTS564) and Heilongjiang Province postdoctoral research start-up Foundation project (2022106).

**Institutional Review Board Statement:** Not applicable.

**Data Availability Statement:** The data that support the findings of this study are available from the corresponding author upon reasonable request.

**Conflicts of Interest:** The authors declare no conflict of interest.

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
