# Peer review of "Effects of Elevated CO2 Concentration and Temperature on the Growth and Photosynthetic Characteristics of Populus simonii × P. nigra ‘1307’ Leaves"

_forests, doi:10.3390/f14112156_

Round 1
Reviewer 1 Report
Comments and Suggestions for Authors
There are several questions and comments about the article.
1) How was the leaf area determined?
2) How was the chlorophyll content calculated?
3) Figures and symbols on them must be made larger.
4) Line 463, 499-501 - change references to numbers.
5) In my opinion, the discussion section needs to be improved, giving additional comparison with literature data.
Author Response
Dear Editor:
Thank you very much for reviewing our manuscript (forests-2598716). We are very grateful to you and the other two reviewers for the helpful suggestions. We have taken the kind suggestions into account and revised the manuscript according to the recommendations. The questions raised by you and reviewers are answered and explained as follows:
Reviewers' comments:Comments to the Author
Reviewer: 1
There are several questions and comments about the article.
- How was the leaf area determined?
Answer: According to the requirements of the reviewer, the mistake has been corrected in the revised MS.
“A scanner (DS-60000, EPSON, Japan) was used to scan Populus spp's leaves, and the leaf area coefficients of 0.75 were obtained through image processing. the leaf area (leaf length x leaf width x 0.75) was also calculated.”
- How was the chlorophyll content calculated?
Answer: According to the requirements of the reviewer, the mistake has been corrected in the revised MS.Line 173-183
According to the OD values of each wavelength, the concentrations of chlorophyll a, b and carotenoids (mg·L-1) were calculated, and the contents of each pigment (mg·g-1 fresh weight) were calculated according to the obtained concentrations. Pigment content in leaves (mg/g)= pigment concentration (mg/L)×total volume of extract (ml)× dilution ratio/sample mass(g). Each measurement (five separate samples) was repeated five times.
Ca=13.95D665-6.88D649
Cb=24.96D649-7.32D665
Cx.c=(1000D470-2.05Ca-114.8Cb)/245
- Figures and symbols on them must be made larger.
Answer: Thank you for your suggestion. We have adjusted the resolution of the image. The reason why the previous version was unclear may be due to the formatting of the MS.
- Line 463, 499-501 - change references to numbers.
Answer: According to the requirements of the reviewer, the mistake has been corrected in the revised MS.
5) In my opinion, the discussion section needs to be improved, giving additional comparison with literature data.
Answer: According to the requirements of the reviewer, the mistake has been corrected in the revised MS.Line381-407; 461-463
Reviewer: 2 Comments to the Author
The presented article is devoted to the revealing of the effects of increased CO2 concentration and temperature on the photosystem II performance and photosynthetic characteristics of Populus. This topic is especially relevant nowadays because of global warming. We need to understand the mechanisms of its influence on our life, including plants growth and development. It is important that increase of CO2 concentration and atmosphere temperature have different effects on plants. It was discovered that under raised temperature conditions, the concentrations of chlorophyll a and b in Populus leaves were significantly increased. But there were no significant changes in the chlorophyll concentration under the superimposed effect of high CO2 concentration and temperature. The continuous increase in temperature by 2°C significantly inhibited the electron transfer capacity of the photosynthetic system in leaves. On the other hand, an increase in CO2 concentration alleviates the photosynthetic inhibition caused by the elevated temperature.
The obtained results indicated complexity of the influence of two main characters of greenhouse effect. Also it shows the necessity of further investigations of plants mechanisms of adaptation.
At the same time I have one remark.
A lot of abbreviations are used in the article and not all of them are decoded. As the article can be useful for many specialists of different specialties, it is better to include the table with the list of abbreviations to the text. At the same time, in the “Abstract” no abbreviations can be used.
Answer: According to the requirements of the reviewer, the mistake has been corrected in the revised MS.Line26.
We will compile all abbreviations and professional vocabulary on the table and upload them as supplementary materials.
Table 1 Formulae and terms used in the analysis of the O-J-I-P fluorescence induction dynamics curve Formulae and terms Illustrations
Fo Minimal recorded fluorescence intensity
Fm Maximal recorded fluorescence intensity
tFm Time to reach maximal fluorescence intensity Fm
VJ Relative variable fluorescence intensity at the J-step
Mo Approximated initial slope of the fluorescence transient
Sm Normalised total complementary area above the O-J-I-P transie (reflecting single-turnover QA reduction events)
Specific energy fluxes [per QA-reducing PSII reaction center (RC)]
ABS/RC Absorption flux per RC
TRo/RC Trapped energy flux per RC (at t=0)
ETo/RC Electron transport flux per RC (at t=0)
DIo/RC Dissipated energy flux per RC (at t=0) Phenomenological energy fluxes [per excited cross section (CS)]
ABS/CSo Absorption flux per CS (at t=0)
TRo/CSo Trapped energy flux per CS (at t=0)
ETo/CSo Electron transport flux per CS (at t=0)
DIo/CSo Dissipated energy flux per CS (at t=0) Density of reaction centers
RC/CSo Density of RCs (QA-reducing PSII reaction centers) Performance indexes
PIABS Performance index on absorption basis
PICS Performance index on cross section basis (at t=0) When t=tFm, CSo was replaced by CSm, ABS/CSm ≈ Fm.
Reviewer: 3 Comments to the Author
This manuscript describes a study to investigate the response photosynthetic physiology, leaf characteristics, and dry matter partitioning mechanism of a Poplar hybrid variety to elevated CO2 concentrations and temperature. The manuscript is in general well written, and should be of some interest to your readers, if the following points can be addressed. My comments and suggestions are as follows:
My main concerns are:
- The study is very short term (4 months). It is not complete. It is misleading as it does not take the plant as a whole and completely avoided the environment.
- It does not account for temperature acclimation.
3.It does not account for drought and precipitation variability, all of which are suggested to be common with increasing CO2.
4.Warming in summer dry periods for example may amplify the effects of intensified summer drought, but promote growth if the plant can survive, in the colder months (early spring and mid-autumn. A potential extension of the growing season.
- It does not factor in the rest of the plant and change in growth, chemistry, and or nutrient cycles within the plant and within the system. Differences in leaf and fine root tissue chemistry may influence feedbacks to carbon and nutrient dynamics through effects on litter decomposition which would affect carbon and nutrient cycles.
- It does not take into consideration seasonal changes!
- Temperature acclimation often results in down regulation of respiration rates with increased plant temperature, thus plant response compared across contrasting growth temperatures may differ from expectations based on short term temperature response function.
- In addition to direct short-term temperature effects, temperature acclimation may modulate net photosynthetic carbon gain and respiratory losses in the leaf and at the plant scale, thereby influencing carbon at the ecosystem scale.
- In addition to physiological acclimation, warming may promote canopy leaf area at the expense of fine root growth or warming induced change in soil moisture may favor the opposite growth portioning pattern. Therefore, the whole plant needs to be considered. Difference in leaf and fine root temperature acclimation may in part, influence growth response.
- Need to think about how A and E may be less responsive to soil water availability and how linked are leaf and fine root traits, and canopy leaf area and fine root standing crop are linked to system nutrient cycles.
Material and methods
Answer: Thank you very much for your suggestion! As of 2020, the total area of poplar plantations in China exceeded 100 million acres, ranking first in the world and making significant contributions to China's forestry projects; Continuing to promote the cultivation of high-quality poplar plantations and researching poplar varieties that can meet socio-economic needs and environmental protection is currently a hot topic in poplar research. This study uses seedlings of hybrid poplar varieties as experimental materials to explore the response mechanism of trees to global change factors such as elevated CO2 concentration and warming from the perspectives of plant photosynthetic physiology and leaf characteristics. Study the effects of climate factors on the physiological characteristics of poplar trees by setting different CO2 concentrations and temperatures in an artificial climate chamber. The research results have important theoretical and practical significance for revealing the adaptive mechanism of trees to environmental changes, as well as for the breeding of poplar artificial forests in northern China and their high-quality development.
Firstly, the reason why we chose poplar seedlings is that the photosynthetic mechanism of the functional leaves of poplar seedlings has been improved, and they are more sensitive to changes in the external environment during the seedling period. Due to the competition between the energy allocation between chlorophyll fluorescence and photosynthesis, it is possible to observe the physiological changes of plants by studying the changes in the amount of fluorescence emitted by plants. This can then infer the growth status of plants and judge their response to the external environment. The kinetic parameters of chlorophyll fluorescence can enrich and indicate the research results. The photosynthesis of trees is the fundamental core of their growth, and the response mechanism of photosynthetic mechanisms to external environmental changes also determines the way trees adapt and regulate their growth in the future. Our experimental results aim to see which part of the photosynthetic system of poplar trees undergoes changes and plays a dominant role in response to external environmental changes. Your suggestions on temperature adaptation, drought caused by temperature rise, seasonal changes, nutrient distribution and cycling within poplar trees, as well as carbon distribution, etc. Some experiments are ongoing research and some will focus on future research directions. While conducting simulation experiments, we will also follow up on outdoor in-situ experiments. Currently, we have actively built OTC chambers to track and test the growth status of poplar seedlings for several consecutive years. Our current research experimental design involves changes in CO2 concentration (isotopes), temperature changes, and drought treatment, and is combined with research on poplar root exudates, litter, and soil microorganisms. Of course, this is a very complex but meaningful research, and your feedback is crucial for our future research. Each question is an inspiration for our future research. Your feedback will greatly help improve our existing experiments and enrich our future experimental design. Thank you again for your valuable feedback!
Line 149-213. Please provide replication number for all work conducted.
Answer: According to the requirements of the reviewer, the mistake has been corrected in the revised MS.
Discussion
Please see above (main concerns). I think the authors d a good job talking about their findings but fails to mention in any details how this work has been done already and has moved on.
Answer: According to the Suggestions given by the reviewer, We have made modifications to our discussion section.
We tried our best to improve the manuscript and made some changes in the manuscript. These changes will not influence the content and framework of the paper. And here we did not list the changes but marked in yellow in revised paper.
We appreciate for Editors and Reviewers’ warm work earnestly, and hope that the correction will meet with approval.
Once again, thank you very much for your comments and suggestions.
Yours sincerely
Nan Xu

Reviewer 2 Report
Comments and Suggestions for Authors
The presented article is devoted to the revealing of the effects of increased CO2 concentration and temperature on the photosystem II performance and photosynthetic characteristics of Populus. This topic is especially relevant nowadays because of global warming. We need to understand the mechanisms of its influence on our life, including plants growth and development. It is important that increase of CO2 concentration and atmosphere temperature have different effects on plants. It was discovered that under raised temperature conditions, the concentrations of chlorophyll a and b in Populus leaves were significantly increased. But there were no significant changes in the chlorophyll concentration under the superimposed effect of high CO2 concentration and temperature. The continuous increase in temperature by 2°C significantly inhibited the electron transfer capacity of the photosynthetic system in leaves. On the other hand, an increase in CO2 concentration alleviates the photosynthetic inhibition caused by the elevated temperature.
The obtained results indicated complexity of the influence of two main characters of greenhouse effect. Also it shows the necessity of further investigations of plants mechanisms of adaptation.
At the same time I have one remark.
A lot of abbreviations are used in the article and not all of them are decoded. As the article can be useful for many specialists of different specialties, it is better to include the table with the list of abbreviations to the text. At the same time, in the “Abstract” no abbreviations can be used.
Author Response

(The authors gave the same response as above.)

Reviewer 3 Report
Comments and Suggestions for Authors
This manuscript describes a study to investigate the response photosynthetic physiology, leaf characteristics, and dry matter partitioning mechanism of a Poplar hybrid variety to elevated CO2 concentrations and temperature. The manuscript is in general well written, and should be of some interest to your readers, if the following points can be addressed. My comments and suggestions are as follows:
My main concerns are:
1. The study is very short term (4 months). It is not complete. It is misleading as it does not take the plant as a whole and completely avoided the environment.
2. It does not account for temperature acclimation.
3. It does not account for drought and precipitation variability, all of which are suggested to be common with increasing CO2.
4. Warming in summer dry periods for example may amplify the effects of intensified summer drought, but promote growth if the plant can survive, in the colder months (early spring and mid-autumn. A potential extension of the growing season.
5. It does not factor in the rest of the plant and change in growth, chemistry, and or nutrient cycles within the plant and within the system. Differences in leaf and fine root tissue chemistry may influence feedbacks to carbon and nutrient dynamics through effects on litter decomposition which would affect carbon and nutrient cycles.
6. It does not take into consideration seasonal changes!
7. Temperature acclimation often results in down regulation of respiration rates with increased plant temperature, thus plant response compared across contrasting growth temperatures may differ from expectations based on short term temperature response function.
8. In addition to direct short-term temperature effects, temperature acclimation may modulate net photosynthetic carbon gain and respiratory losses in the leaf and at the plant scale, thereby influencing carbon at the ecosystem scale.
9. In addition to physiological acclimation, warming may promote canopy leaf area at the expense of fine root growth or warming induced change in soil moisture may favor the opposite growth portioning pattern. Therefore, the whole plant needs to be considered. Difference in leaf and fine root temperature acclimation may in part, influence growth response.
10. Need to think about how A and E may be less responsive to soil water availability and how linked are leaf and fine root traits, and canopy leaf area and fine root standing crop are linked to system nutrient cycles.
Material and methods
Line 149-213. Please provide replication number for all work conducted.
Discussion
Please see above (main concerns). I think the authors d a good job talking about their findings but fails to mention in any details how this work has been done already and has moved on.
Author Response

(The authors gave the same response as above.)

Round 2
Reviewer 3 Report
Comments and Suggestions for Authors
1. The study is very short term (4 months). It is not complete. It is misleading as it does not take the plant as a whole and completely avoided the environment.
2. It does not account for temperature acclimation.
3. It does not account for drought and precipitation variability, all of which are suggested to be common with increasing CO2.
4.Warming in summer dry periods for example may amplify the effects of intensified summer drought, but promote growth if the plant can survive, in the colder months (early spring and mid-autumn. A potential extension of the growing season.
5. It does not factor in the rest of the plant and change in growth, chemistry, and or nutrient cycles within the plant and within the system. Differences in leaf and fine root tissue chemistry may influence feedbacks to carbon and nutrient dynamics through effects on litter decomposition which would affect carbon and nutrient cycles.
6. It does not take into consideration seasonal changes! 7. Temperature acclimation often results in down regulation of respiration rates with increased plant temperature, thus plant response compared across contrasting growth temperatures may differ from expectations based on short term temperature response function.
8. In addition to direct short-term temperature effects, temperature acclimation may modulate net photosynthetic carbon gain and respiratory losses in the leaf and at the plant scale, thereby influencing carbon at the ecosystem scale.
9. In addition to physiological acclimation, warming may promote canopy leaf area at the expense of fine root growth or warming induced change in soil moisture may favor the opposite growth portioning pattern. Therefore, the whole plant needs to be considered. Difference in leaf and fine root temperature acclimation may in part, influence growth response.
10. Need to think about how A and E may be less responsive to soil water availability and how linked are leaf and fine root traits, and canopy leaf area and fine root standing crop are linked to system nutrient cycles.
Author Response
Dear Editor:
Thank you very much for reviewing our manuscript (forests-2598716). We are very grateful to you and the other two reviewers for the helpful suggestions. We have taken the kind suggestions into account and revised the manuscript according to the recommendations. The questions raised by you and reviewers are answered and explained as follows:
Reviewers' comments:Comments to the Author
Reviewer: 1
There are several questions and comments about the article.
- How was the leaf area determined?
Answer: According to the requirements of the reviewer, the mistake has been corrected in the revised MS.
“A scanner (DS-60000, EPSON, Japan) was used to scan Populus spp's leaves, and the leaf area coefficients of 0.75 were obtained through image processing. the leaf area (leaf length x leaf width x 0.75) was also calculated.”
- How was the chlorophyll content calculated?
Answer: According to the requirements of the reviewer, the mistake has been corrected in the revised MS.Line 173-183
According to the OD values of each wavelength, the concentrations of chlorophyll a, b and carotenoids (mg·L-1) were calculated, and the contents of each pigment (mg·g-1 fresh weight) were calculated according to the obtained concentrations. Pigment content in leaves (mg/g)= pigment concentration (mg/L)×total volume of extract (ml)× dilution ratio/sample mass(g). Each measurement (five separate samples) was repeated five times.
Ca=13.95D665-6.88D649
Cb=24.96D649-7.32D665
Cx.c=(1000D470-2.05Ca-114.8Cb)/245
- Figures and symbols on them must be made larger.
Answer: Thank you for your suggestion. We have adjusted the resolution of the image. The reason why the previous version was unclear may be due to the formatting of the MS.
- Line 463, 499-501 - change references to numbers.
Answer: According to the requirements of the reviewer, the mistake has been corrected in the revised MS.
5) In my opinion, the discussion section needs to be improved, giving additional comparison with literature data.
Answer: According to the requirements of the reviewer, the mistake has been corrected in the revised MS.Line381-407; 461-463
Reviewer: 2 Comments to the Author
The presented article is devoted to the revealing of the effects of increased CO2 concentration and temperature on the photosystem II performance and photosynthetic characteristics of Populus. This topic is especially relevant nowadays because of global warming. We need to understand the mechanisms of its influence on our life, including plants growth and development. It is important that increase of CO2 concentration and atmosphere temperature have different effects on plants. It was discovered that under raised temperature conditions, the concentrations of chlorophyll a and b in Populus leaves were significantly increased. But there were no significant changes in the chlorophyll concentration under the superimposed effect of high CO2 concentration and temperature. The continuous increase in temperature by 2°C significantly inhibited the electron transfer capacity of the photosynthetic system in leaves. On the other hand, an increase in CO2 concentration alleviates the photosynthetic inhibition caused by the elevated temperature.
The obtained results indicated complexity of the influence of two main characters of greenhouse effect. Also it shows the necessity of further investigations of plants mechanisms of adaptation.
At the same time I have one remark.
A lot of abbreviations are used in the article and not all of them are decoded. As the article can be useful for many specialists of different specialties, it is better to include the table with the list of abbreviations to the text. At the same time, in the “Abstract” no abbreviations can be used.
Answer: According to the requirements of the reviewer, the mistake has been corrected in the revised MS.Line26.
We will compile all abbreviations and professional vocabulary on the table and upload them as supplementary materials.
Table 1 Formulae and terms used in the analysis of the O-J-I-P fluorescence induction dynamics curve Formulae and terms Illustrations
Fo Minimal recorded fluorescence intensity
Fm Maximal recorded fluorescence intensity
tFm Time to reach maximal fluorescence intensity Fm
VJ Relative variable fluorescence intensity at the J-step
Mo Approximated initial slope of the fluorescence transient
Sm Normalised total complementary area above the O-J-I-P transie (reflecting single-turnover QA reduction events)
Specific energy fluxes [per QA-reducing PSII reaction center (RC)]
ABS/RC Absorption flux per RC
TRo/RC Trapped energy flux per RC (at t=0)
ETo/RC Electron transport flux per RC (at t=0)
DIo/RC Dissipated energy flux per RC (at t=0) Phenomenological energy fluxes [per excited cross section (CS)]
ABS/CSo Absorption flux per CS (at t=0)
TRo/CSo Trapped energy flux per CS (at t=0)
ETo/CSo Electron transport flux per CS (at t=0)
DIo/CSo Dissipated energy flux per CS (at t=0) Density of reaction centers
RC/CSo Density of RCs (QA-reducing PSII reaction centers) Performance indexes
PIABS Performance index on absorption basis
PICS Performance index on cross section basis (at t=0) When t=tFm, CSo was replaced by CSm, ABS/CSm ≈ Fm.
Reviewer: 3 Comments to the Author
This manuscript describes a study to investigate the response photosynthetic physiology, leaf characteristics, and dry matter partitioning mechanism of a Poplar hybrid variety to elevated CO2 concentrations and temperature. The manuscript is in general well written, and should be of some interest to your readers, if the following points can be addressed. My comments and suggestions are as follows:
My main concerns are:
Answer: Firstly, the reason why we chose poplar seedlings is that the photosynthetic mechanism of the functional leaves of poplar seedlings has been improved, and they are more sensitive to changes in the external environment during the seedling period. Due to the competition between the energy allocation between chlorophyll fluorescence and photosynthesis, it is possible to observe the physiological changes of plants by studying the changes in the amount of fluorescence emitted by plants. This can then infer the growth status of plants and judge their response to the external environment. The kinetic parameters of chlorophyll fluorescence can enrich and indicate the research results. The photosynthesis of trees is the fundamental core of their growth, and the response mechanism of photosynthetic mechanisms to external environmental changes also determines the way trees adapt and regulate their growth in the future.
Our experimental results aim to see which part of the photosynthetic system of poplar trees undergoes changes and plays a dominant role in response to external environmental changes. Your suggestions on temperature adaptation, drought caused by temperature rise, seasonal changes, nutrient distribution and cycling within poplar trees, as well as carbon distribution, etc. Some experiments are ongoing research and some will focus on future research directions. While conducting simulation experiments, we will also follow up on outdoor in-situ experiments. Currently, we have actively built OTC chambers to track and test the growth status of poplar seedlings for several consecutive years. Our current research experimental design involves changes in CO2 concentration (isotopes), temperature changes, and drought treatment, and is combined with research on poplar root exudates, litter, and soil microorganisms. Of course, this is a very complex but meaningful research, and your feedback is crucial for our future research. Each question is an inspiration for our future research. Your feedback will greatly help improve our existing experiments and enrich our future experimental design. Thank you again for your valuable feedback!
1.The study is very short term (4 months). It is not complete. It is misleading as it does not take the plant as a whole and completely avoided the environment.
Answer: Thank you very much for your suggestion. The reason why we choose a short test period is to adapt and adjust the photosynthetic system function of saplings in the early growth stage and the plant growth state to the changes in the external environment, that is to say, the initial response of plants to the external environment and the way they respond. The key factor determining plant growth is photosynthesis, which responds quickly to the changes in the external environment. So that's one of the reasons for our study, and we have similar studies on that.
“Xu N#, Zhang HH#, Zhong HX, et al. The response of photosynthetic functions of F1 cutting seedlings from Physocarpus amurensis Maxim (♀) × Physocarpus opulifolius “Diabolo” (♂) and the parental leaves to salt stress. Frontiers in Plant Science. 2018,9:714”
Zhang HH, Feng P, Yang W, Sui X, Li X, Zhang RT, Gu SY, Xu N. Effects of flooding stress on the photosynthetic apparatus of leaves of two Physocarpus cultivars. Journal of Forest Research, 2018,39(4):1049-1059
Xu N, Zhang HH, Sui X, et al, Wang JF, Qu Y, Sun GY. Arbuscular mycorrhizal fungi (Glomus mosseae) improves growth, photosynthesis and protects photosystem II in leaves of Lolium perenne L. under cadmium contaminated soil. Frontiers in Plant Science.2018,9:1156
In addition, we also have corresponding experimental designs for long-term photosynthesis and growth monitoring, and your suggestions are very helpful to us.
- It does not account for temperature acclimation.
Answer: Thank you very much for your suggestion. Our research aims to adapt and adjust the photosynthetic system function of saplings in the early growth stage and the growth state of plants to changes in the external environment, that is to say, the initial response of plants to the external environment and the way of response. The key factor determining plant growth is photosynthesis, so we designed this experiment in response to changes in the external environment. This paper mainly studies the physiological aspects of plants, especially the photosynthetic mechanism. We've done similar research on temperature acclimation.
“Xu N, Liu XJ, Zhang HH, et al.Chilling acclimation and the response of antioxidant enzymes to chilling stress in mulberry seedlings. Journal of Forest Research, 2018,39(4):1049-1059”
From this point of view, we will provide a very targeted research direction for the relevant experiments on the response of trees to long-term environmental changes
3.It does not account for drought and precipitation variability, all of which are suggested to be common with increasing CO2.
Answer: Thank you for your advice. Drought and precipitation will affect the growth of plants, and there is a certain correlation with the increase of carbon dioxide in the external environment, but this is a complicated process. The changes in the external environment simulated by our experiment are mainly temperature and carbon dioxide concentration, and other experimental conditions are kept under certain experimental conditions as far as possible. Under the same conditions for each experimental treatment, the impact on growth and photosynthetic mechanism is mainly reflected in the external temperature and carbon dioxide concentration changes. We have had similar studies. In field in-situ experiments and indoor simulation experiments, plants are adaptive to long-term environmental changes, but their respective adjustment methods are not the same, but the initial performance is from the photosynthetic apparatus, determining the growth and accumulation and transport of organic matter, and even affecting the leaf structure of new leaves, such as stomata, palisade tissue and so on. These are a regulatory effect to adapt to the external environment.
4.Warming in summer dry periods for example may amplify the effects of intensified summer drought, but promote growth if the plant can survive, in the colder months (early spring and mid-autumn. A potential extension of the growing season.
Answer: Thank you for your suggestion, which is related to the first three questions. We also have relevant long-term monitoring experiments, including field in-situ experiments and indoor simulation experiments. Plants are adaptable to long-term external environmental changes, but their adjustment methods are not the same. It even affects the leaf structure of the new leaves, such as stomata, palisade tissue and so on. These are a regulatory effect to adapt to the external environment.
- It does not factor in the rest of the plant and change in growth, chemistry, and or nutrient cycles within the plant and within the system. Differences in leaf and fine root tissue chemistry may influence feedbacks to carbon and nutrient dynamics through effects on litter decomposition which would affect carbon and nutrient cycles.
Answer: Thank you for your suggestion. Just like some of our studies, plants are adaptive to long-term environmental changes, but their regulation methods are not the same, but the initial performance starts from the photosynthetic apparatus, determines the growth and accumulation and transport of organic matter, and even affects the leaf structure of the new leaves, such as stomata, palisade tissue and so on. These are a regulatory effect to adapt to the external environment. For example, we have done a preliminary study that different environments change the growth of taproot or lateral roots of plants, as well as root secretions, etc., and the structure of different roots and different secretions also have certain effects on the microorganisms in the soil where the plants are located. We fully agree with your suggestion, and we will give full consideration to the experiments already conducted and the design of future experiments.
6.It does not take into consideration seasonal changes!
Answer: Thank you for your suggestion. As you said, the change of seasons will affect the growth environment of plants. Our research mainly focuses on the influence of plants on the changes of external environmental factors (increasing temperature and carbon dioxide concentration) and their photosynthetic organs, as well as how plants respond to the photosynthetic organs and growth, which will be instructive for our long-term monitoring in the later stage.
- Temperature acclimation often results in down regulation of respiration rates with increased plant temperature, thus plant response compared across contrasting growth temperatures may differ from expectations based on short term temperature response function.
Answer: Thank you for your suggestion. As you said, the change of seasons will affect the growing environment of plants. Our research mainly focuses on the influence of plants on the changes of external environmental factors (temperature increase) and their photosynthetic organs, as well as how the photosynthetic organs and growth of plants respond, and mainly focuses on the response changes of individual plants. There will be different expressions of the results.
- In addition to direct short-term temperature effects, temperature acclimation may modulate net photosynthetic carbon gain and respiratory losses in the leaf and at the plant scale, thereby influencing carbon at the ecosystem scale.
Answer:Thank you for your suggestion. As you said, the change of seasons will affect the growing environment of plants. Our research mainly focuses on the influence of plants on the changes of external environmental factors (temperature increase) and their photosynthetic organs, as well as how the photosynthetic organs and growth of plants respond, and mainly focuses on the response changes of individual plants. There will be different expressions of the results. The fixation of carbon in the whole ecosystem is a goal of our future experiments. We report that we have also conducted relevant experimental studies using isotopes. The impact of external environment changes on plant carbon allocation will be fully adopted in our future large-scale experiments.
9.In addition to physiological acclimation, warming may promote canopy leaf area at the expense of fine root growth or warming induced change in soil moisture may favor the opposite growth portioning pattern. Therefore, the whole plant needs to be considered. Difference in leaf and fine root temperature acclimation may in part, influence growth response.
Answer: Thank you for your suggestion. For example, we have done a preliminary study that different environments change the growth of taproot or lateral roots of plants, as well as root secretions, etc., and the structure of different roots and different secretions also have certain effects on the microorganisms in the soil where the plants are located. We fully agree with your suggestion, and we will give full consideration to the experiments already conducted and the design of future experiments.
- Need to think about how A and E may be less responsive to soil water availability and how linked are leaf and fine root traits, and canopy leaf area and fine root standing crop are linked to system nutrient cycles.
Material and methods
Answer: Thank you very much for your suggestion! As of 2020, the total area of poplar plantations in China exceeded 100 million acres, ranking first in the world and making significant contributions to China's forestry projects; Continuing to promote the cultivation of high-quality poplar plantations and researching poplar varieties that can meet socio-economic needs and environmental protection is currently a hot topic in poplar research. This study uses seedlings of hybrid poplar varieties as experimental materials to explore the response mechanism of trees to global change factors such as elevated CO2 concentration and warming from the perspectives of plant photosynthetic physiology and leaf characteristics. Study the effects of climate factors on the physiological characteristics of poplar trees by setting different CO2 concentrations and temperatures in an artificial climate chamber. The research results have important theoretical and practical significance for revealing the adaptive mechanism of trees to environmental changes, as well as for the breeding of poplar artificial forests in northern China and their high-quality development.
Line 149-213. Please provide replication number for all work conducted.
Answer: According to the requirements of the reviewer, the mistake has been corrected in the revised MS.
Discussion
Please see above (main concerns). I think the authors d a good job talking about their findings but fails to mention in any details how this work has been done already and has moved on.
Answer: According to the Suggestions given by the reviewer, We have made modifications to our discussion section.
We tried our best to improve the manuscript and made some changes in the manuscript. These changes will not influence the content and framework of the paper. And here we did not list the changes but marked in yellow in revised paper.
We appreciate for Editors and Reviewers’ warm work earnestly, and hope that the correction will meet with approval.
Once again, thank you very much for your comments and suggestions.
Yours sincerely
Nan Xu
